# *Breynia cernua*: Chemical Profiling of Volatile Compounds in the Stem Extract and Its Antioxidant, Antibacterial, Antiplasmodial and Anticancer Activity In Vitro and In Silico

**DOI:** 10.3390/metabo13020281

**Published:** 2023-02-15

**Authors:** Hesti Lina Wiraswati, Nisa Fauziah, Gita Widya Pradini, Dikdik Kurnia, Reza Abdul Kodir, Afiat Berbudi, Annisa Retno Arimdayu, Amila Laelalugina, Ilma Fauziah Ma’ruf

**Affiliations:** 1Department of Biomedical Sciences, Faculty of Medicine, Universitas Padjadjaran, Sumedang 45363, Indonesia; 2Advance Biomedical Laboratory, Faculty of Medicine, Universitas Padjadjaran, Bandung 40161, Indonesia; 3Infection Working Group, Faculty of Medicine, Universitas Padjadjaran, Sumedang 45363, Indonesia; 4Oncology and Stem Cell Working Group, Faculty of Medicine, Universitas Padjadjaran, Bandung 40161, Indonesia; 5Departement of Chemistry, Faculty of Mathematics and Natural Sciences, Universitas Padjadjaran, Sumedang 45363, Indonesia; 6Pharmacy Study Program, Faculty of Mathematics and Natural Sciences, Universitas Islam Bandung, Bandung 40116, Indonesia; 7PT. Borneo Indobara, Central Jakarta 10350, Indonesia; 8Department of Mining Engineering, Faculty of Technology Mineral, Institut Teknologi Nasional Yogyakarta, Yogyakarta 55281, Indonesia; 9Biochemistry Research Group, Department of Chemistry, Faculty of Mathematics and Natural Sciences, Institut Teknologi Bandung, Bandung 40132, Indonesia

**Keywords:** *B. cernua* (stem) methanol extract, metabolite profiling, antioxidant, antibacterial, antiplasmodial, anticancer, molecular docking, drug likeness, pharmacokinetics

## Abstract

*Breynia cernua* has been used as an alternative medicine for wounds, smallpox, cervical cancer, and breast cancer. This plant is a potential source of new plant-derived drugs to cure numerous diseases for its multiple therapeutic functions. An in vitro study revealed that the methanol extract of *B. cernua* (stem) exhibits antioxidant activity according to DPPH and SOD methods, with IC_50_ values of 33 and 8.13 ppm, respectively. The extract also exerts antibacterial activity against *Staphylococcus aureus* with minimum bactericidal concentration of 1875 ppm. Further analysis revealed that the extract with a concentration of 1–2 ppm protects erythrocytes from the ring formation stage of *Plasmodium falciparum*, while the extract with a concentration of 1600 ppm induced apoptosis in the MCF-7 breast cancer cell line. GC–MS analysis showed 45 bioactive compounds consisting of cyclic, alkyl halide, organosulfur, and organoarsenic compounds. Virtual screening via a blind docking approach was conducted to analyze the binding affinity of each metabolite against various target proteins. The results unveiled that two compounds, namely, N-[β-hydroxy-β-[4-[1-adamantyl-6,8-dichloro]quinolyl]ethyl]piperidine and 1,3-phenylene, bis(3-phenylpropenoate), demonstrated the best binding score toward four tested proteins with a binding affinity varying from −8.3 to −10.8 kcal/mol. Site-specific docking analysis showed that the two compounds showed similar binding energy with native ligands. This finding indicated that the two phenolic compounds could be novel antioxidant, antibacterial, antiplasmodial, and anticancer drugs. A thorough analysis by monitoring drug likeness and pharmacokinetics revealed that almost all the identified compounds can be considered as drugs, and they have good solubility, oral bioavailability, and synthetic accessibility. Altogether, the in vitro and in silico analysis suggested that the extract of *B.* cernua (stem) contains various compounds that might be correlated with its bioactivities.

## 1. Introduction

From a historical point of view, medicinal plants have been proven to have therapeutic properties to cure various diseases, and they still represent an important source of novel drugs today [1,2]. More than 50% of drugs in modern therapeutics are represented by plant-derived drugs, either in their natural or in their derivative form [3,4]. For example, successful plant-derived drugs are aspirin from *Filipendula ulmaria* (L.) Maxim and artemisinin from *Artemisia annua* L. [5,6]. Indonesia is a country with high plant biodiversity and has very diverse plant resources (approximately 30,000–40,000 plant species), 20% of which are either wild or cultivated medicinal plants [7].

On the other hand, metabolite profiling and the exploration of medicinal plants in Indonesia are still rarely conducted. For those reasons, the metabolite profile of uninvestigated plants in Indonesia is interesting to explore since they are a promising source of new plant-derived drugs. Therefore, this research aimed to unveil the metabolite component of one herbaceous plant as a reservoir of a particular bioactive compound that can potentially be used as a drug in the future.

Various *Breynia* species have been traditionally applied as medicine for numerous diseases [8]. One such species, *Breynia cernua* (local name: Katuk Hutan), has been used in Amuntai, Hulu Sungai Utara, and Kalimantan Selatan as herbal medicine for smallpox and wounds [9], as well as in Papua as a traditional medicine for cervical and breast cancer [10]. *B. cernua* has the following anatomical characteristics: semilunar vascular bundle, paracyitic and anomocytic stomata, irregular shape of the lower epidermis with a wavy anticlinal wall, polygonal shape of the upper epidermis with a straight anticlinal wall, single unicellular trichome on the lower epidermis, one-layer upper epidermis with a regularly shaped cell, one-layer lower epidermis with a papillated cell, one-layer elongated palisade tissue, loose arrangement of the spongy tissue, druse calcium oxalate on vascular bundle parenchyma, and hypodermis in the area near leaf bone [11]. It was found that the ethanol extract of *B. cernua* has cytotoxic activity according to the brine shrimp lethality test and MTT assay against MCF-7 [10]. The ethyl acetate or butanol extract of *B. cernua* (stem hardwood) and methanol extract of *B. cernua* (stem bark) exhibited broad-spectrum antibacterial activities against *Micrococcus luteus*, *Micrococcus roseus, Bacillus coagulans, Bacillus cereus, Bacillus megaterium, Bacillus subtilis, Citrobacter freundii, Lactobacillus casei, Staphylococcus albus, Staphylococcus aureus, Staphylococcus epidermidis, Agrobacterium tumefaciens, Streptococcus faecalis, Streptococcus pneumoniae, Klebsiella pneumoniae, Eschericia coli, Proteus vulgaris, Proteus mirabili, Neisseria gonorrhoeae, Pseudomonas aeruginosa, Enterobacter aerogenes, Salmonella* Typhi, *Salmonella* Typhimurium, and *Serratia marcescens*. Furthermore, the butanol or methanol extract of *B. cernua* (root bark) demonstrated antifungal activities against *Aspergillus niger, Aspergillus versicolour, Aspergillus vitis, Candida albican, Candida tropicalis, Cladosporium cladosporides, Penicillium notatum, Trichophyton mentagrophyte, Tricophyton rubrum*, and *Tricophyton tonsurans* [12]. The ethanol extract of *B. cernua* also was found to have antiparkinsonism activity in vitro and in vivo [13]. Considering its multiple in vivo and in vitro bioactivity, *B. cernua* is a promising source of novel plant-derived drugs. Therefore, identification of its chemical composition and activity prediction of each metabolite represent interesting research pathways.

Traditional medicinal plant research has gradually risen worldwide in recent years, owing to the natural sources and variety of such plants, which allow them to complement current pharmaceutical treatments [14,15]. The bioactivities of medicinal plants are correlated with the phytochemicals contained in them such as alkaloids, flavonoids, saponins, terpenoids, tannins, and quinonens [16]. Metabolite profiling through GC–MS using the silyl derivatization method has been widely applied to identify the composition of plant extracts [17,18]. In addition, bioinformatics tools aid drug discovery by using all primary data gathered from in vivo and in vitro studies in a short time and at a lower cost. In silico molecular docking is one of the most effective ways to find novel ligands for proteins with known structures, thus playing an essential role in structure-based drug development [19]. A preliminary study (GC–MS) showed that the stem extract of *B. cernua* contained more compounds than the leaf extract. Moreover, the stem extract exhibited better antioxidant and bactericidal activity than the leaf extract (manuscript in preparation). Therefore, in this report, we performed metabolite profiling, characterized the in vitro antioxidant, antibacterial, antiplasmodial, and anticancer activity, and conducted virtual drug screening through molecular docking, drug likeness, and pharmacokinetic analyses of the methanol extract of *B. cernua* (stem).

## 2. Materials and Methods

### 2.1. Plant Material and Preparation of the Extracts

*Breynia cernua* was collected from a coal mining reclamation area at Site Kusan-Girimulya of PT Borneo Indobara. The plant material’s taxonomic identification was validated by reference books and research articles [20]. To obtain the extract, 200 g of *B. cernua* stems were cut and oven-dried (50 °C, 48 h) before being macerated in 100 mL of methanol (24 h, room temperature). The solvent was removed by oven-drying (70 °C, 4 h), and the extract was solubilized using DMSO (0.5% *v*/*v*).

### 2.2. Gas Chromatography/Mass Spectrometry (GC–MS) Method

GC–MS was used to identify compounds in the methanol extract of *B. cernua* (stem). Sample derivatization was conducted using the trimethyl silyl derivatization (TMS) method. Briefly, 20 µL of extract was dried using an oven (80 °C, 20 min). Immediately after, the sample was solubilized using 80 µL of methoxyamine hydrochloride solution in pyridine (2 mg/100 mL), mixed by vortexing for 1 min, and incubated for 90 min at 30 °C. Next, the solubilized sample was combined with 80 µL of MSTFA and incubated for 30 min at 37 °C prior to GC–MS analysis [21]. GC–MS analyses of *B. cernua* (stem) extracts were carried out using an Agilent 5977B GC/MS (made in USA). Pure helium gas was employed as the carrier gas (1 mL/min), and the injector temperature was set at 270 °C. The oven temperature program was set as follows: gradual increase from 70 °C to 200 °C (10 °C/min), then to 310 °C (10 °C/min), before holding at 310 °C for 5 min. The mass spectrometer was set to electron ionization mode at 70 eV with an electron multiplier voltage of 1859 V. The retention index of compounds was recorded using Mass Hunter GC/MS Acquisition 10.0.368 software. The phytochemical content determination was conducted by comparing the results to authentic compound spectral databases stored in the National Institute of Standards and Technology (NIST) library.

### 2.3. Antioxidant Activity Test

The antioxidant activity test was conducted using DPPH and SOD methods. The assay was repeated with solvent (DMSO 0.5% *v*/*v*) as a negative control.

#### 2.3.1. DPPH Method

First, 4 × 10^−4^ M DPPH solution was prepared and placed into a vial, closed tightly and wrapped using aluminium foil. From the stock solution (1000 ppm), the sample concentration was varied from 0.5 to 42 ppm and placed in test tubes. After that, 1 mL of DPPH solution was added to each test tube and allowed to sit for 30 min, followed by absorbance measurements (λ517 nm). The following equation was used to calculate the ability to scavenge DPPH free radicals (inhibition): %h = (Ab – As)/Ab × 100%, where %h is the percentage inhibition (free-radical inhibition), Ab is the absorbance blank, and As is the sample absorbance. The value of 50% free-radical inhibition concentration (IC_50_) was calculated using the regression equation y = ax + b.

#### 2.3.2. Method for Superoxide Dismutase (SOD) Activity Assay

From the stock solution (1000 ppm), the sample concentration was varied from 5 to 20 ppm and placed in test tubes. The SOD activity was measured using a xanthine–xanthine oxidase system capable of generating superoxide radicals. At λ550 nm, the degree of suppression of nitro blue tetrazolium reduction by O_2_ was measured. The amount of enzyme required to inhibit nitro blue tetrazolium degradation by 50% in 1 min was measured as U/mg protein [22]. The following equation was used to calculate the percentage inhibition: %h = (Ab – As)/Ab × 100%, where %h is the percentage inhibition (free-radical inhibition), Ab is the absorbance blank, and As is the sample absorbance. The value of 50% free-radical inhibition concentration (IC_50_) was calculated using the regression equation y = ax + b.

### 2.4. Minimum Inhibitory Concentration against Staphylococcus aureus

The broth microdilution method was used to determine the minimum inhibitory concentration (MIC) according to CLSI [23]. The extracts were serially diluted twice in a microtiter plate with Mueller–Hinton broth. Each plate received a bacterial inoculum with a final concentration of 5 × 10^5^ CFU/mL and was incubated at 37 °C for 24 h. The MIC was defined as the lowest extract’s concentration that inhibited bacterial growth. The minimum bactericidal concentration (MBC) was considered as the lowest extract concentration resulting in no bacterial growth. Treatment with solvent (DMSO 0.5% *v*/*v*) was also used as a negative control.

### 2.5. Antiplasmodial Assay

*P. falciparum* strain FCR-3/Gambia was purchased from ATCC (30932^TM^). The cells were grown in Roswell Park Memorial Institute’s medium (RPMI 1640 R8578) supplemented with inactivated blood type O serum in a 9:1 ratio. Serum was extracted from whole blood and centrifuged for 10 min at 1600 rpm. The serum was then inactivated by heating it at 56 °C for 30 min. *P. falciparum* cells were cultivated in a complete medium in red blood cells (RBCs) at 37 °C and 5% CO_2_. The medium was regularly replaced twice a week. After 1 week of maintenance (all RBCs were lysed by parasites), the cells were ready to be used for the next experiment. The red blood cells (RBCs) were obtained from blood type O, which was centrifuged at 1600 rpm for 10 min. The pellet of RBC was washed in phosphate-buffered saline solution and resuspended with complete medium in a ratio of 1:1. For treatment, infected RBCs were prepared by adding 1 × 10^6^ RBCs to complete medium and 10 µL of a *P. falciparum* cell suspension in a 50 mL Falcon tube. Extracts of *B. cernua* at various concentrations (600 ppm, 300 ppm, 150 ppm, 100 ppm, 50 ppm, 20 ppm, 10 ppm, 5 ppm, 2 ppm, or 1 ppm) were administered to the same Falcon tube. Treatment with solvent (DMSO 0.5% *v*/*v*) was also used as a negative control. Suspended cells were grown for 2 × 24 h on six-well plates at 37 °C and 5% CO_2_. The positive control consisted of RBCs and *P. falciparum*, while the negative control consisted of RBCs. Then, cells were harvested to calculate the number of intact RBCs using a hemocytometer, and the infected RBCs were observed using Giemsa staining. Giemsa staining was applied by placing 10 µL of cell suspension on a slide, fixing it with methanol, and incubating it with Giemsa solution for 10 min. The presence of *P. falciparum* in RBCs was determined using a light microscope.

### 2.6. Cytotoxic Activity

The MCF-7 human breast cancer cell line was purchased from the ATCC (HTB-22^TM^) (transcriptome sequence data were deposited in the ArrayExpress database https://www.ebi.ac.uk/biostudies/arrayexpress/ (accessed on 21 May 2022) by Chiang et al. with accession number E-MTAB-609) [24]. The cells were grown in Eagle’s minimum essential medium (EMEM) supplemented with 1% penicillin/streptomycin and 10% heat-inactivated fetal bovine serum in a 96-well plate (37 °C, 5% CO_2_, 20 h). Then, the cells were treated with various concentrations of *B. cernua* stem extract (200, 400, 800, and 1600 ppm) for 3, 6, 8, and 24 h. Treatment with solvent (DMSO 0.5% *v*/*v*) was also used as a negative control. The cell viability was estimated by measuring absorbance at λ570 nm using a Quant ELISA plate reader (Agilent, CA, USA). Trypan blue staining was used to measure the level of cell death. The morphology of cells was observed using inverted microscopy (Agilent).

### 2.7. Molecular Modeling (Receptors and Ligand Preparation)

The crystal structure of the selected holoenzymes was downloaded from the RCSB Protein Databank (PDB): [25] 3EUB [26], 3VSL [27], 3QS1 [28], and 3GD4 [29]. Water and ligand molecules present in the pdb files were removed using UCSF Chimera [30]. Ligands were built and energy-minimized (Dreiding force field) using MarvinSketch software https://chemaxon.com/marvin (accessed on 8 May 2022).

### 2.8. Molecular Docking

Virtual screening was conducted using PyrX software [31] and Vina software version 1.2.3. [32,33] using a blind docking approach. Specific docking was conducted using UCSF Chimera software [30] and Vina software version 1.2.3. [32,33]. For the specific docking process, the grid box of each protein was set as follows: cefotaxime-binding site on 3VSL (center_x = 19.00, center_y = −50.00, center_z = 24.00, size_x = 17.00, size_y = 17.00, size_z = 17.00); FAD-binding site on 3EUB (center_x = −17.00, center_y = 13.00, center_z = −84.00, size_x = 19.00, size_y = 19.00, size_z = 19.00); MTE-binding site on 3EUB (center_x = −42.00, center_y = 18.00, center_z = −55.00, size_x = 15.00, size_y = 15.00, size_z = 15.00); KNI-10006-binding site on 3QS1 (center_x = 28.00, center_y = −9.00, center_z = 5.00, size_x = 19.00, size_y = 17.00, size_z = 17.00); FAD-binding site on 3GD4 (center_x = −39.00, center_y = 31.00, center_z = −66.00, size_x = 20.00, size_y = 22.00, size_z = 20.00); NAD-binding site on 3GD4 (center_x = −50.00, center_y = 35.00, center_z = −62.00, size_x = 16.00, size_y = 15.00, size_z = 16.00). The best docking results were chosen and visualized using Biovia Discovery Studio software (https://3ds.com/products-services/biovia/products) (accessed on 8 May 2022).

### 2.9. Drug Likeness and Pharmakokinetic Analysis

Drug likeness and pharmacokinetic analyses of molecules were analyzed using the SwissADME server (http://www.swissadme.ch/)(accessed on 8 May 2022). The drug likeness analysis was conducted according to the criteria of Lipinski, Ghose, Veber, Egan, and Muegge.

## 3. Results and Discussion

### 3.1. Metabolite Profilling

Phytochemical analysis of leaf extract of several *Breynia* species such as *B. coronata, B. fruticosa, B. retusa, B. androgyna, B. glauca B. officinalis, B. rostrata,* and *B. vitis-idaea* was conducted. The identified compounds from several Breynia species were grouped into alkaloids, aromatic ketones, catechins, flavonoids, glycosides, lignans, neolignans, steroids, terpenoids, and tannins [8]. Metabolite profiling through GC–MS analysis demonstrated the presence of 45 compounds in the methanol extract of *B. cernua* (stem) (Table 1) which could contribute to the therapeutic effects of this plant. Seven of them were phenolic compounds: N-[β-hydroxy-β-[4-[1-adamantyl-6,8-dichloro]quinolyl]ethyl]piperidine; propanoic acid, 2,2-dimethyl-, 2-(1,1-dimethylethyl)phenyl ester; benzenepropanoic acid, α-hydroxy-, methyl ester; 1,3-phenylene, bis(3-phenylpropenoate); benzenepropanoic acid, 3,5-bis(1,1-dimethylethyl)-4-hydroxy-, methyl ester; 1,4-bis(trimethylsilyl)benzene; 4-methyl-2,4-bis(p-hydroxyphenyl)pent-1-ene, 2TMS derivative. In addition, there were 10 cyclic compounds: oxalic acid, cyclohexyl nonyl ester; cyclohexene, 2-ethenyl-1,3,3-trimethyl-; 1,2:4,5:9,10-triepoxydecane; 1,3-cyclohexadiene, 5-(1,5-dimethyl-4-hexenyl)-2-methyl-, [S-(R*,S*)]-; 2-piperidinone, N-[4-bromo-n-butyl]-; ethylamine, 2-(adamantan-1-yl)-1-methyl-; methyl 3-bromo-1-adamantaneacetate; 1-cyclohexylnonene; cyclohexene, 1-nonyl-; cyclotrisiloxane, hexamethyl-. Furthermore, there were alkyl halides (methane, dichloronitro-; 1-chloroundecane; dodecane, 1-chloro-; decane 1-chloro-; tetradecane, 1-chloro-; 1-octadecanesulphonyl chloride; 2-bromotetradecane; 2-bromo dodecane; dodecane, 1-iodo-; eicosane, 1-iodo-), three organosulfur compounds (sulfurous acid, hexyl nonyl ester; sulfurous acid, decyl hexyl ester; sulfurous acid, 2-ethylhexyl isohexyl ester), and one organoarsen compound (arsenous acid, tris(trimethylsilyl) ester). Unfortunately, compounds with many hydroxyl groups such as flavonoids and phenolic acids were not detected in this extract. Therefore, this study focused on the volatile compounds of *B. cernua* (stem).

Medicinal plants are a promising reservoir of bioactive metabolites that can support the discovery of novel drug molecules. GC–MS analysis was conducted on the *B. cernua* (stem) methanol extract and revealed 45 bioactive compounds. Studies in the literature have revealed the isolation of several metabolites, regardless of whether their specific function was demonstrated or not. Meanwhile, the origin or bioactivity of a few metabolites has not been documented. Dichloronitromethane, as the dominant compound in ethyl acetate fraction of endophytic fungus *Alternaria alternata* AE1, exhibited antimicrobial properties [34]. 3,3,6-Trimethyl-1,5-heptadien-4-ol identified, as one of the main compounds of *Artemisia austro-yunnanensis* essential oil, possessed antioxidant activity [35]. Sulfurous acid, 2-ethylhexyl isohexyl ester, contained in the methanol extract of *Gracilaria corticata* and *Gracilaria edulis*, showed antioxidant and antibacterial activity [36]. Sulfurous acid, hexyl nonyl ester was identified in the ethanol extract of *Eclipta prostrata* and showed antimicrobial activity [37]. Decane, 2,4-dimethyl- was identified in the endophytic bacterium *Pantoea* sp. strain Dez632 and possessed antibacterial activity [38]. Octane, 3,4,5,6-tetramethyl-, contained in the endophytic bacterium *Pseudomonas* sp. strain Bt851, exhibited antibacterial activity [38]. Oxalic acid, cyclohexyl nonyl ester, isolated from the methanol extract of *Rheum ribes*, showed an inhibitory effect on *Escherichia coli* [39]. Octane, 2,4,6-trimethyl- was identified in the essential oil of *Rumex hastatus*, which exhibited inhibitory and antioxidant activity [40]. 1-Chloroundecane, contained in the extract of *Streptomyces* sp., exerted antibacterial activity [41]. Dodecane, 1-chloro- was identified in the extract of *Spongia officinalis* var *ceylonensis* and possessed antiproliferative activity against three cancer cell lines [42]. Decane, 1-chloro-, identified in the ethyl acetate fraction of *Artocarpus anisophyllus* Miq stem bark, exerted antioxidant activity [43]. Sulfurous acid, decyl hexyl ester was identified in rambutan seed fat [44] and as a volatile organic compound in olive tree [45]. Tetradecane, 1-chloro- was identified in the crude extract of medicinal herbs used to treat kidney stones [46]. Dodecane, 4,6-dimethyl-, contained in the essential oil of *Ephedra pachyclada*, possessed antioxidant and antimicrobial activity [47]. Cyclohexene, 2-ethenyl-1,3,3-trimethyl-, extracted from the ethyl acetate fraction of *Streptomyces* sp. Al-Dhabi-90 metabolites, demonstrated broad-spectrum activity against pathogenic bacteria [48]. Hexadecane was identified as a major component of *Monochaetia kansensis* chloroform extract, and this compound is known to have antioxidant and antibacterial activity [49]. Benzenepropanoic acid, α-hydroxy-, methyl ester (synonym: lactic acid, 3-phenyl-, methyl ester), contained in *Lactobacillus plantarum* culture supernatants, possessed potential antipathogenic and chronic skin wound-healing properties [50]. 1,3-Phenylene, bis(3-phenylpropenoate), identified in the acetone extract of *Jasminum annamense* subsp. *annamense* leaves, exerted antioxidant and antibacterial activity [51]. 1,2:4,5:9,10-Triepoxydecane was extracted from the ethyl acetate fraction of *Alysicarpus glumaceus*, which had therapeutic properties such as stimulatory, diuretic, antipsychotic, anti-inflammatory, antidiarrheal, abortifacient, antitussive, and anti-asthmatic activity [52]. 1,3-Cyclohexadiene, 5-(1,5-dimethyl-4-hexenyl)-2-methyl-, [S-(R*,S*)]- was identified in the methanol extract of *Zingiber officinale*, and this compound had pharmacological properties such as antioxidant, anti-inflammatory, and antinociceptive activities [53]. 1-Octadecanesulphonyl chloride was previously isolated from the volatile floral extract of *Acacia auriculiformis,* which exhibited antioxidant and antifungal activity [54]. Oxalic acid, allyl pentadecyl ester was identified in the methanol extract of red alga *Laurencia brandenii*, which showed anti-pest, antimicrobial, termiticidal, and maggoticidal activity [55]. 2-Piperidinone, N-[4-bromo-n-butyl]- was contained in various organic solvent extracts of pomegranate peel and demonstrated antimicrobial activities against *P. aeruginosa*, *P. mirabilis*, and *C. albicans* [56]. 2-Bromotetradecane, as one of the major constituents of ethyl acetate fraction of *Haematocarpus validus* Bakh. F. Ex Forman, exhibited antimicrobial and antioxidant activity [57]. Oxalic acid, 6-ethyloct-3-yl heptyl ester, as one of the major constituents of *Ricinus communis* seed oil, exhibited anti-inflammatory, antibacterial, antioxidant, anthelmintic, antidiabetic, anticancer, mosquitocidal, and insecticidal activity [58]. 2-Bromododecane was identified in the n-hexane extract of *Moringa oleifera* root bark and exhibited antibacterial activity [59]. Ethylamine, 2-(adamantan-1-yl)-1-methyl- was contained in the essential oil from *Cinnamomum tamala* dried leaves, and this plant has been reported to have therapeutic activities to treat anorexia, shore throat, skin diseases, colds, cough, colic, and diarrhea [60]. Pentadecanal, isolated from *Pseudoalteromonas haloplanktis* TAC125 culture supernatant, exhibited inhibitory activity toward the biofilm formation of *S. epidermidis* [61]. Di-n-decylsulfone was identified in the acetone extract of *Spatholobus purpureus* root, and this plant has been used to cure animal hemorrhagic septicemia [62]. Dodecane-1-iodo- was present in *Rhazya stricta chloroform extract,* which exhibited antidiabetic activity [63]. Hexadecanoic acid, methyl ester, contained in the FAME extract of *Sesuvium portulacastrum*, possessed antibacterial, anticandidal, and antifungal activity [64]. Benzenepropanoic acid, 3,5-bis(1,1-dimethylethyl)-4-hydroxy-, methyl ester, present in the hexane, ethyl acetate, and methanol extracts of *Ficus bhotanica*, had bioactivities such as antifungal and antioxidant properties [65]. Methyl 3-bromo-1-adamantaneacetate, as the major component of *Caulerpa racemosa*, possessed antibacterial and larvicidal activity [66]. 1-Cyclohexylnonene was identified in the n-butanol extract of *Ehretia serrata* and exhibited broad-spectrum antimicrobial activity [67]. 4-Methyl-2,4-bis(p-hydroxyphenyl)pent-1-ene, 2TMS derivative was identified in the aqueous extract of *Luffa acutangula* peel [68], and this compound has been known to have anticancer activity by inducing apoptosis of pancreatic β-cells [69]. Cyclohexene, 1-nonyl- was identified as a volatile compound in *Piper guineense* leaves and seeds, and this spice exerts medicinal functions such as for the treatment of rheumatism, bronchitis, cough, and intestinal disease. Moreover, the leaf extract exhibited antimicrobial properties [70]. Cyclotrisiloxane, hexamethyl-, mainly contained in the acetone extract of *Turbinaria decurrens*, exhibited antioxidant and antidiabetic activity [71]. 1,4-Bis(trimethylsilyl)benzene was found in the volatile oil of *Glycosmis pentaphylla*, and this plant has in vivo bioactivities such as antidiabetic, antipyretic, antioxidant, antibacterial, anthelmintic, antinociceptive, and hepatoprotective properties [72]. 9,12,15-Octadecatrienoic acid, methyl ester, (Z,Z,Z)- was found in the n-hexane extract of *Archidium ohioense* (Schimp. ex Mull), and this compound has been known to have cardioprotective, antioxidant, anticancer, antipyretic, antibacterial, antiarthritic, and antiandrogenic activities [73]. Eicosane, 1-iodo-, contained in the ethyl acetate and petroleum ether extracts of *Alnus cremastogyne* pods, exhibited antioxidant activity [74]. Arsenous acid, tris(trimethylsilyl) ester is one of the major compounds of *Momordica cymbalaria* methanol extract, and this plant is commonly used to cure rheumatism, skin disease, diabetes mellitus, diarrhea, and ulcers [75]. From the above evidence, we can conclude that the *B. cernua* extract (stem) contains numerous compounds with various potential bioactivities.

### 3.2. Antioxidant and Antibacterial Test Results

The extract exhibited remarkable antioxidant activity as measured using the DPPH (2,2-diphenyl-1-pikrilhidrazil) and SOD (superoxide dismutase) methods, with IC_50_ values of 33 and 8.13 ppm, respectively. Numerous phenolic compounds can result in antioxidant activity since the phenolic and flavonoid groups commonly act as antioxidants, negating the oxidative damage generated by free radicals at the molecular level [76]. In addition, the extract also possessed bactericidal activity against pathogenic bacterium *S. aureus* at a concentration of 1875 ppm. This finding is in line with a prior study on *B. cernua* extract, which exhibited broad-spectrum antimicrobial activity against bacteria, including *S. aureus* [12].

### 3.3. Antiplasmodial Test Result

An antimalarial test was conducted on *Plasmodium falciparum* FBR3. The test parameters were the number of intact erythrocytes and the parasitemia level of the treatment using both extracts at various concentrations with an incubation period of 2 × 24 h, compared to the negative control. The results demonstrated that the methanol extract was active against *P. falciparum*. Both methanol extract treatments (1 and 2 ppm) show good activity. The number of erythrocytes decreased on *P. falciparum*-infected erythrocytes; however, the 1–2 ppm extract treatment protected cells from lysis (Figure 1 and Figure 2). At 10 ppm, the extract seemingly did not have a protective effect since the number of intact erythrocytes was similar to the control (+). Further analysis showed that the methanol extract might play a role in protecting erythrocytes from ring formation stage of *P. falciparum* infection (Figure 3).

### 3.4. Anticancer Test Result

Previous research stated that n-hexane fraction of *B. cernua* had the highest cytotoxic toward MCF-7 breast cancer cells compared to ethyl acetate and water fractions, with an IC_50_ value of 165.65 ppm. On the basis of these data, a test was conducted on the methanol extract of *B. cernua* using the MCF-7 cell line. The test was conducted at various concentrations to determine the extract’s activity in killing cancer cells. The result showed that the extract started killing cancer cells within 3 h of treatment at concentrations of 1600 ppm and 800 ppm. Concentrations under 800 ppm did not show any effects on the cells. (Figure 4). Further morphology analysis showed that *B. cernua* (1600 ppm, 3 h) induced cell shrinkage and apoptotic body formation, which are hallmarks of apoptosis (Figure 5).

### 3.5. Virtual Screening with Blind Docking Approach

Virtual screening was used to estimate each compound’s affinity to selected target proteins: 3VSL as an antibacterial protein model, 3EUB as an antioxidant protein model, 3QS1 as an antiplasmodial protein model, and 3GD4 as an anticancer protein model. Blind docking performed using Pyrx software revealed that 44 of 45 compounds were successfully docked into the target proteins. However, the docking analysis could not be performed using arsenous acid, tris(trimethylsilyl) because the arsenic atom could not be recognized by AutoDock Vina software (Table 2).

*B. cernua* is known as an alternative medicine with antiviral properties toward smallpox [9] and anticancer properties toward breast and cervical cancer [10]. In addition, its methanol extract also exerts broad-spectrum antibacterial activity [12], as well as antioxidant, antiplasmodial, and anticancer properties (this study). Computational docking tools are used by researchers all around the world to uncover and evaluate the binding affinity of compounds that fit a protein’s binding site [19]. From the above in vitro and in vivo evidence, four protein models were chosen as molecular docking targets, as they were previously crystallized and/or used for in silico studies to investigate the binding affinity of particular compounds: 3EUB as an antioxidant target protein [77], 3VSL as an antibacterial target protein [78], 3QS1 as an antiplasmodial target protein **[79],** and 3GD4 as an anticancer target protein [80]. According to the virtual screening results (Table 2), N-[β-hydroxy-β-[4-[1-adamantyl-6,8-dichloro]quinolyl]ethyl]piperidine and 1,3-phenylene, bis(3-phenylpropenoate) exhibited the best binding affinity toward all protein models. Hence, the detailed interactions between the two phenolic compounds and the target proteins were further investigated.

Xanthine oxidase (pdb:3EUB) was selected as an antioxidant protein target since the methanol extract of *B. cernua* stem possessed inhibitory activity against the enzyme according to the SOD assay.The binding interaction between 3EUB and N-[β-hydroxy-β-[4-[1-adamantyl-6,8-dichloro]quinolyl]ethyl]piperidine revealed one conventional hydrogen bond formed between Leu257 and the oxygen atom of the ligand. One electrostatic interaction was formed between the ligand and Lys256 on 3EUB (π–cation). Furthermore, there were 16 hydrophobic interactions involving Ile264 (alkyl), Leu257, Ile353, Lys395, Leu398 (π–alkyl), and 11 other amino acids including Arg394 (van der Waals) (Figure 6). The binding interaction between 3EUB and 1,3-phenylene, bis(3-phenylpropenoate) revealed two conventional hydrogen bonds formed between Leu257 and Val259 and the carboxyl group of the ligand. Meanwhile, a van der Waals interaction was formed between Lys256 and 1,3-phenylene, bis(3-phenylpropenoate). Moreover, there were 16 hydrophobic interactions involving Ala301 (π–alkyl), Leu257 (π–sigma), and 14 other amino acids (van der Waals) (Appendix A). Arg880 and Glu1261 are active residues of xanthine oxidase [26]. Both plant-derived molecules could be antioxidant drugs by binding to xanthine oxidase in a competitive manner, leading to the reduction in reactive oxygen species in the cell.

Penicillin-binding protein 3 (PBP3) from methicilin-resistant *S. aureus* (pdb:3VSL) was selected as an antibacterial target since the methanol extract of *B. cernua* stem exhibited bactericidal activity against pathogenic bacteria. The binding interaction between 3VSL and N-[β-hydroxy-β-[4-[1-adamantyl-6,8-dichloro]quinolyl]ethyl]piperidine revealed two conventional hydrogen bonds formed between Asn487 and Gly492 and the oxygen atom of ligand. Furthermore, there were 17 hydrophobic interactions involving Pro253, Leu256, Lys273 (alkyl), Tyr278, Tyr275, Ile381, Ile507 (π–alkyl), and 10 other amino acids (van der Waals) (Appendix A). The binding interaction between 3VSL and 1,3-phenylene, bis(3-phenylpropenoate) revealed three conventional hydrogen bonds formed between Tyr275, Arg483, and Arg504 and the carboxyl groups of the ligand. One electrostatic interaction was formed between the ligand and Arg483 on 3VSL (π–cation). Moreover, there were 11 hydrophobic interactions involving Ile381 (π–alkyl), Arg483 (π–sigma), and nine other amino acids (van der Waals) (Figure 7). Ser392, Ser393, Val394, Lys395, Ser448, Ser449, Asn450, Lys608, Thre609, and Gly610 are active residues of methicillin-resistant *S. aureus* penicillin-binding protein 3/PBP3 [27]. Both plant-derived molecules could be promising antibacterial drugs by binding to PBP3 in a noncompetitive manner, leading to the inhibition of bacterial cell-wall formation.

Plasmepsin I (PMI) from *P. falciparum* (pdb:3QS1) was selected as an antiplasmodial target since the methanol extract of *B. cernua* (stem) was proven to inhibit parasite growth, as indicated by healthy erythrocytes (Figure 3). The binding interaction between 3QS1 and N-[β-hydroxy-β-[4-[1-adamantyl-6,8-dichloro]quinolyl]ethyl]piperidine only revealed hydrophobic interactions such as Ile287 (alkyl), Val12, Ile30, Ala111, Phe117, Ile287, Pro343 (π–alkyl), and 14 other amino acids including Ser77, Phe109, Ile120, Thr218, Ser219, Thr222, and Phe242 (Van der Waals) (Appendix A). Ile30, Ile287, Ser77, Phe109, Ile120, Thr218, Ser219, Thr222, and Phe242 are part of the Plasmepsin 1-binding site [28]. The binding interaction between 3QS1 and 1,3-phenylene, bis(3-phenylpropenoate) revealed only hydrophobic interactions: Ile30, Pro110, Ala111, Leu243, (π–alkyl), Phe117 (π–sigma), and eight other amino acids including Phe109, Ile120, and Phe242 (van der Waals) (Appendix A). Ile30, Phe109, Ile120, and Phe242 are part of the Plasmepsin 1-binding site [28]. Both molecules could be promising antiplasmodial drug candidates by inhibiting *P. falciparum* Plasmepsin I, leading to inhibition of haemoglobin degradation.

Apoptosis induction factor (AIF) was selected as an anticancer target protein since the methanol extract of *B. cernua* (stem) induced apoptotic cell death of the MCF-7 cell line (Figure 4 and Figure 5). The binding interaction between AIF (3GD4) and N-[β-hydroxy-β-[4-[1-adamantyl-6,8-dichloro]quinolyl]ethyl] piperidine revealed the following hydrophobic interactions: Phe583 (alkyl) Pro172, Phe192, Trp482 (π–alkyl), Trp482 (π–π stacked), and 11 other amino acids including Lys176 (van der Waals) (Appendix A). Interestingly, Trp482, Pro172, and Lys 176 are part of the quinone (menadione)-binding site [80], and this molecule has been proven to induce apoptosis in cancer cells via thiodione formation mediated by AIF [81]. N-[β-Hydroxy-β-[4-[1-adamantyl-6,8-dichloro]quinolyl]ethyl]piperidine is a quinone derivative that could be a new anticancer drug candidate with a mechanism of action by inducing apoptosis in cancer cells. The binding interaction between 3GD4 and 1,3-phenylene, bis(3-phenylpropenoate) revealed one conventional hydrogen bond formed between Arg568 and the carboxyl group of the ligand. In addition, there was one π–donor hydrogen bond (Arg448) and one electrostatic (π–cation) interaction between the ligand and Arg449. Moreover, there were 15 hydrophobic interactions involving Val451, Leu501 (π–alkyl), Leu296 (π–sigma), and 12 other amino acids (van der Waals) (Appendix A).

### 3.6. Site-Specific Docking on Binding Site of Native Ligands

Site-specific docking was performed to compare the binding energy of N-[β-hydroxy-β-[4-[1-adamantyl-6,8-dichloro]quinolyl]ethyl]piperidine and 1,3-phenylene, bis(3-phenylpropenoate) with the native ligands of each protein. Table 3 reveals that the two compounds exhibited almost similar binding energy to the native ligands of each protein, except for the binding energy of N-[β-hydroxy-β-[4-[1-adamantyl-6,8-dichloro]quinolyl]ethyl]piperidine on MTE binding site of 3EUB, which was much lower than that of the native ligand. Appendix A visualize the binding interactions between the native ligands and the isolated compounds and each target protein. The result demonstrates that the two compounds have potential application as novel drugs for the inhibition of antibacterial, antioxidant, antiplasmodial, and anticancer protein targets in a competitive manner. Nevertheless, an experimental investigation is needed to validate these findings. Therefore, the next step will be to obtain these compounds naturally or synthetically.

### 3.7. Drug Likeness and Pharmacokinetic Analyses

The qualification of each identified compound as an oral drug was conducted according to the criteria of Lipinski, Ghose, Veber, Egan, and Muegge (Table 4). Lipinski’s rule of five is as follows: molecular weight < 500 Da, <5 hydrogen bond donors, <10 hydrogen bond acceptors, and log P < 5. Ghose utilizes four parameters to qualify a molecule as drug: 160 Da ≤ molecular weight ≤ 480 Da, −0.4 ≤ log P ≤ 5.6, 40 ≤ omlar refractivity ≤ 130, and 20 ≤ total atom number ≤ 70. Veber bases drug likeness on two parameters: topological polar surface area (TPSA) < 140 A^2^ and number of rotatable bonds <10. Egan considers three criteria: TPSA <132 A^2^ and −1 ≤ log P ≤6. Muegge establishes drug likeness as follows: 200 Da ≤ molecular weight ≤ 400 Da, −2 ≤ log P ≤ 5, TPSA ≤ 150 A^2^, <5 hydrogen bond donors, <10 hydrogen bond acceptors, <15 rotatable bonds, <7 rings, and at least four carbon atoms and one heteroatom. Solubility was measured using the ESOL model, where a log S value ≤−6 indicates a poorly soluble compound, while a log S value ≤−10 indicates an insoluble compound. Additionally, a compound with 0.55 ≤ bioavailability score ≤ 0.85 and 0.25 ≤ Csp3 (degree of flexibility) score ≤1 is qualified as orally bioavailable [82]. Compounds with no more than one violation are promising drug candidates. According to Lipinski’s criteria, all of the identified molecules in the methanol extract of *B. cernua* (stem) can be considered oral drug candidates. According to Ghose’s criteria, almost all molecules can be considered drug candidates except for compounds 1 and 10. The filter process using Veber and Egan criteria showed that all molecules qualified as drug candidates. On the basis of Muegge’s criteria, only compounds 4, 5, 8, 10, 14, 21, 22, 23, 27, 35, 36, 37, 39, and 41 qualified as drug candidates. The bioavailability and Csp3 score revealed that all compounds were orally bioavailable except compound 22.

The pharmacokinetic parameters of each compound (Table 5) were predicted in terms of gastrointestinal absorption (GI), blood–brain barrier (BBB) permeability, and skin permeability (log Kp). A more negative log Kp indicates lower skin permeability of the molecule. Another factor underlying pharmacokinetics is the interaction between the molecule and particular proteins such as permeability glycoprotein (P-gp) and cytochromes P450 (CYPs). P-gp is a key factor in active efflux toward the biological membrane, while CYPs are proteins that play a role in drug biotransformation. The synthetic accessibility score ranges from 1 (very easy) to 10 (very difficult) [83]. Table 5 shows that half of the compounds had high GI absorption, whereas the other half had low GI absorption. To solve this challenge, scientists have been using particular strategies such as encapsulating drugs in lipid formulations [84]. The pharmacokinetic analysis also revealed that half of the compounds exhibited low penetration into the BBB and half of the compounds exhibited high penetration into the BBB. Generally, a low penetration of drugs into the BBB is required to minimize the side-effects on the central nervous system (CNS). On the other hand, drugs with high permeability across the BBB are needed to cure cancer patients with brain metastases [85]. The prediction of the interactions with pharmacokinetic proteins revealed that almost all compounds had a low interaction with CYP enzymes, thus indicating minimal drug–drug interactions. Moreover, all compounds had easy to moderate synthetic accessibility. In view of this thorough analysis, these compounds can be good drug leads of natural origin. Further studies on the purification or synthesis of these can be conducted to reveal the mechanisms underlying their antioxidant, antibacterial, antiplasmodial, and anticancer activities.

## 4. Conclusions

As a country with huge biodiversity, Indonesia has abundant plant resources to be used as herbal medicine or as metabolite reservoirs for drug discovery. Amongst them, *B. cernua* is a potential medicinal plant to be explored since it has been traditionally used to treat various diseases. We provided in vitro evidence that the methanol extract of *B. cernua* (stem) exhibits antioxidant, antibacterial, antiplasmodial, and anticancer activities. In addition, the plant has been traditionally demonstrated to have antiviral activity. Further analysis revealed that the extract induced apoptosis in the MCF-7 breast cancer cell line and protected erythrocytes from the ring formation stage of *P. falciparum.* Then, to identify important phytoconstituents, the extract’s metabolites were profiled using GC–MS analysis, which revealed 45 compounds that could contribute to the plant’s therapeutic functions. Bioinformatics was used for drug discovery to save time and chemical reagents. Therefore, blind docking was performed to identify the binding affinity of the metabolites toward target proteins, which revealed that N-[β-hydroxy-β-[4-[1-adamantyl-6,8-dichloro]quinolyl]ethyl]piperidine and 1,3-phenylene, bis(3-phenylpropenoate) exhibited the best results. Further analysis showed that the two molecules were predicted to have an inhibition effect toward antioxidant, antiplasmodial, and anticancer protein models in a competitive manner. Meanwhile, the two molecules were predicted to have an inhibitory effect toward the antibacterial protein model in a noncompetitive manner. Site-specific docking analysis showed that the two compounds exhibited similar binding energy to the native ligands. Interestingly, their bioactivities have not yet been documented. The drug likeness and pharmacokinetic analyses revealed that almost all identified compounds can be considered promising drugs, showing good oral bioavailability, high absorption, and good synthetic availability. Our results clearly indicate that the in silico studies were supported by in vitro studies.

## Figures and Tables

**Figure 1 metabolites-13-00281-f001:**
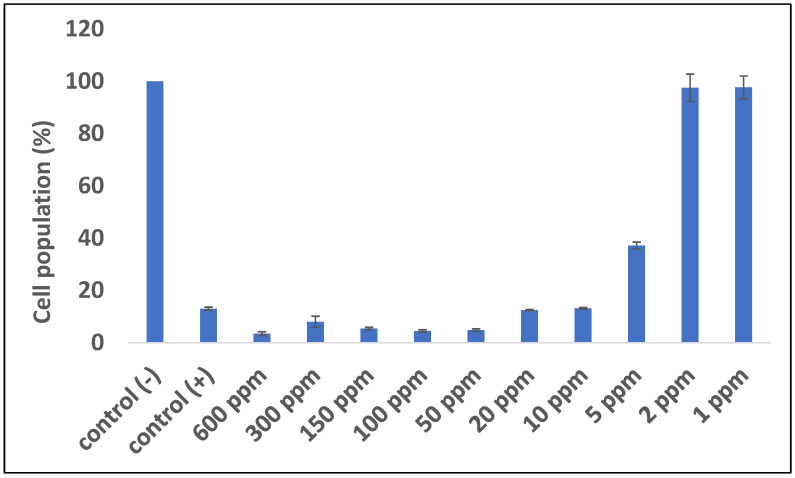
Quantitative analysis of the impact of *B. cernua* extract addition to malarial parasite-infected erythrocytes. Control (−): erythrocytes; control (+): erythrocytes + *P. falciparum*; treatment: erythrocytes *+ P. falciparum* + methanol extracts at various concentrations (600 ppm, 300 ppm, 150 ppm, 100 ppm, 50 ppm, 20 ppm, 10 ppm, 5 ppm, 2 ppm, or 1 ppm). All of the data except for 1 and 2 ppm extract were considered significantly differ compared to the control (−) (*p* <0.05).

**Figure 2 metabolites-13-00281-f002:**
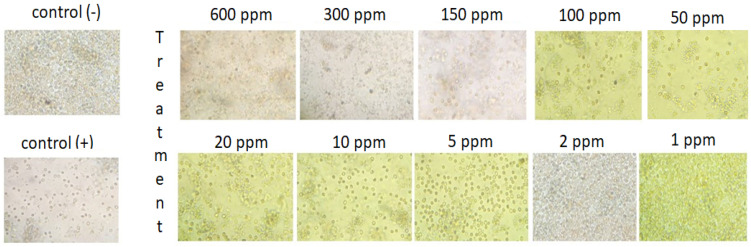
Qualitative analysis of the impact of *B. cernua* extract addition to malarial parasite-infected erythrocytes. Control (−): erythrocytes; control (+): erythrocytes + *P. falciparum*; treatment: erythrocytes *+ P. falciparum* + methanol extracts at various concentrations (600 ppm, 300 ppm, 150 ppm, 100 ppm, 50 ppm, 20 ppm, 10 ppm, 5 ppm, 2 ppm, or 1 ppm).

**Figure 3 metabolites-13-00281-f003:**
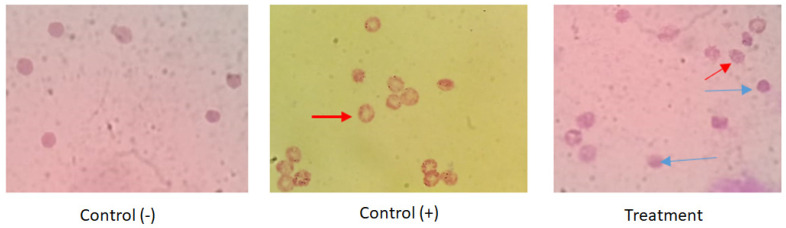
The impact of *B. cernua* extract addition to the malarial parasite-infected erythrocytes (Giemsa staining). Control (−): erythrocytes; control (+): erythrocytes + *P. falciparum*; treatment: erythrocytes *+ P. falciparum* + methanol extract at 5 ppm. Red arrow: infected erythrocytes at the ring formation stage; blue arrow: intact cells following extract treatment.

**Figure 4 metabolites-13-00281-f004:**
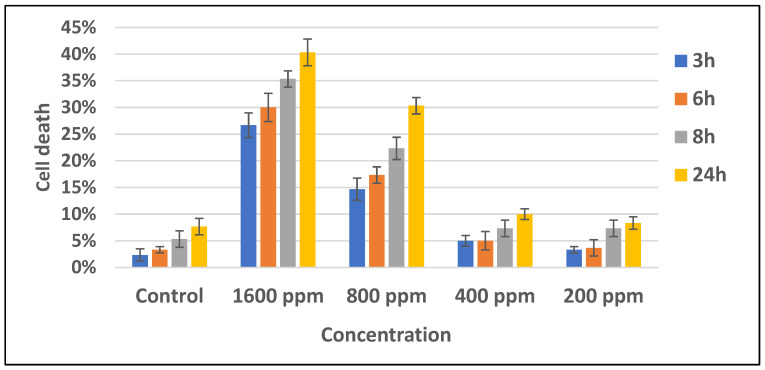
*Brenia cernua* induced the death of MCF-7 cancer cells (dose and time dependency). The cell viability was estimated by measuring the absorbance at λ570 nm using a Quant ELISA plate reader (Agilent BioTek Instruments). Control: cancer cells treated with solvent (DMSO 0.5% *v*/*v*).

**Figure 5 metabolites-13-00281-f005:**
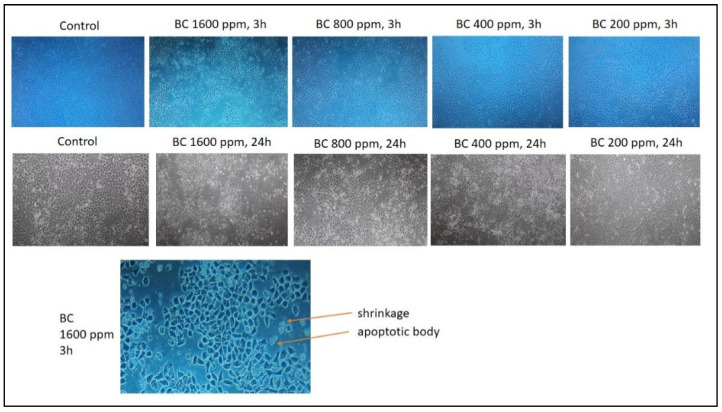
The application of *Brenia cernua* to MCF-7 cell line. Trypan blue staining was used to measure the level of cell death. Control: cancer cells treated with solvent (DMSO 0.5% *v*/*v*). The cell morphology was observed using inverted microscopy (Agilent). Apoptotic bodies were characterized by cell shrinkage.

**Figure 6 metabolites-13-00281-f006:**
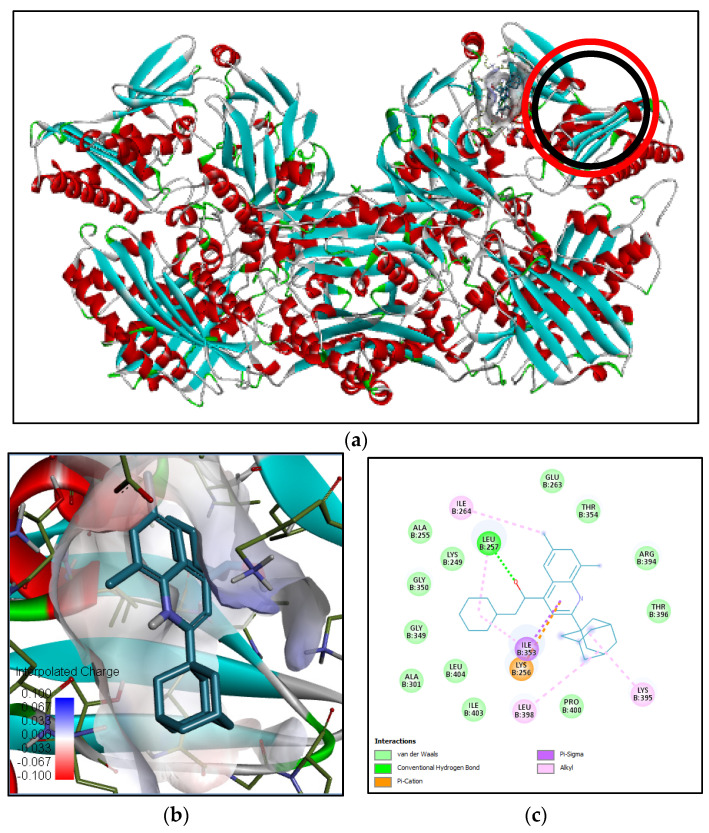
Binding interaction of N-[β-hydroxy-β-[4-[1-adamantyl-6,8-dichloro]quinolyl]ethyl] piperidine compound with antioxidant target protein (3EUB) based on the binding energy generated by PyRx program: (**a**) a close-up view of ligand binding on 3EUB (the binding sites of the native ligand and docked ligand are indicated by red and black circles, respectively); (**b**) 3D diagram of ligand–protein binding pose; (**c**) 2D illustration of ligand–protein interactions.

**Figure 7 metabolites-13-00281-f007:**
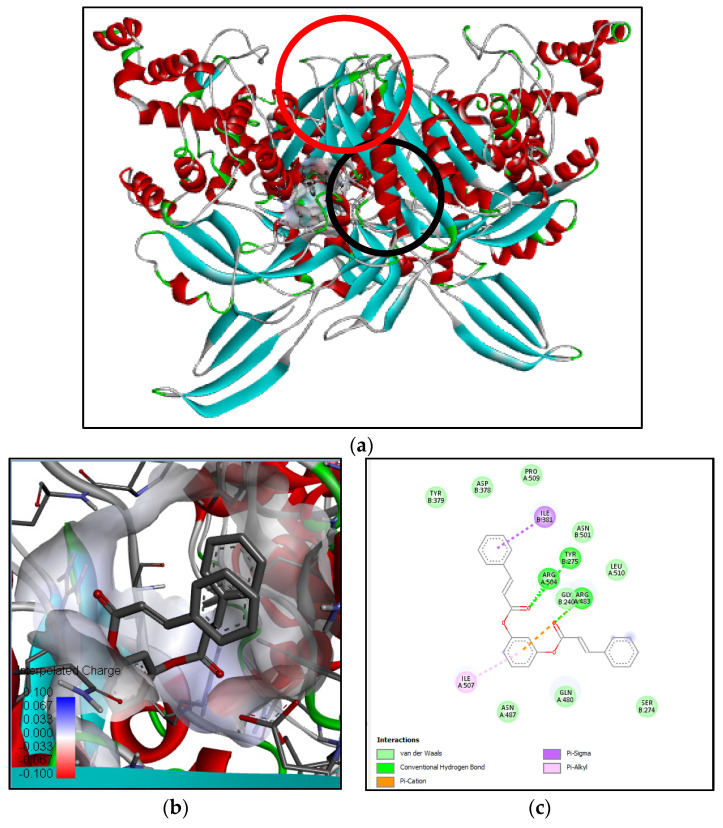
Binding interaction of 1,3-phenylene, bis(3-phenylpropenoate) compound with antibacterial target protein (3VSL) based on the binding energy generated by PyRx program: (**a**) a close-up view of ligand binding on 3VSL (binding sites of the native ligand and docked ligand are indicated by red and black circles, respectively); (**b**) 3D diagram of ligand–protein binding pose; (**c**) 2D illustration of ligand–protein interactions.

**Table 1 metabolites-13-00281-t001:** Metabolite profiling of *B. cernua* methanol extract using gas chromatography/mass spectrometry (GC–MS) method.

No	Compound Name	Retention Time (minute)	Molecular Formula	*m*/*z*
1	Methane, dichloronitro-	3322–3386	CHCl_2_NO_2_	43, 54, 68, 82, 90, 134
2	Ether, hexyl pentyl	4135–4187	C_11_H_24_O	42, 84, 127
3	1,5-Heptadien-4-ol, 3,3,6-trimethyl-	7410–7453	C_10_H_18_O	42, 56, 70, 82, 97
4	Sulfurous acid, 2-ethylhexyl isohexyl ester	7574–7612	C_14_H_30_O_3_S	42, 56, 82, 90, 112
5	Sulfurous acid, hexyl nonyl ester	8008–8046	C_15_H_32_O_3_S	42, 56, 76, 82, 94
6	Decane, 2,4-dimethyl-	8119–8156	C_12_H_26_	43, 84, 112, 126
7	Octane, 3,4,5,6-tetramethyl-	8213–8269	C_12_H_26_	42, 56, 70, 82, 90, 112, 126
8	Oxalic acid, cyclohexyl nonyl ester	8558–8593	C_17_H_30_O_4_	42, 56, 68, 82, 94, 110
9	Octane, 2,4,6-trimethyl-	8857–8903	C_11_H_24_	42, 56, 70, 84, 90, 112, 126
10	N-[β-Hydroxy-β-[4-[1-adamantyl-6,8-dichloro]quinolyl]ethyl]piperidine	9383–9.400	C_27_H_33_Cl_2_NO	43, 54, 69, 97, 118, 134, 149
11	1-Chloroundecane	10,582–10,614	C_11_H_23_Cl	42, 56, 68, 84, 104, 118, 134, 149
12	Dodecane, 1-chloro-	10,628–10,660	C_12_H_25_Cl	42, 56, 70, 84, 104, 134
13	Decane, 1-chloro-	10,760–10,803	C_10_H_21_Cl	42, 56, 68, 84, 104, 134, 149
14	Propanoic acid, 2,2-dimethyl-, 2-(1,1-dimethylethyl) phenyl ester	10,897–10,948	C_15_H_22_O_2_	43, 56, 78, 90, 106, 118, 134, 149
15	Sulfurous acid, decyl hexyl ester	11,347–11,377	C_16_H_34_O_3_S	42, 84, 98, 112, 134, 154
16	Tetradecane, 1-chloro-	11,436–11,479	C_14_H_29_Cl	42, 56, 70, 84, 104, 118, 134, 149
17	Dodecane, 4,6-dimethyl-	11,522–11,557	C_14_H_30_	43, 71, 98, 126, 154
18	Cyclohexene, 2-ethenyl-1,3,3-trimethyl-	1894–11,975	C_14_H_30_	42, 56, 70, 90, 104, 118, 134, 149
19	Pentane, 2,3,3-trimethyl-	12,112–12,134	C_8_H_18_	42, 56, 70, 84, 104, 118, 134, 149, 168
20	Hexadecane	12,182–12,220	C_16_H_34_	42, 56, 84, 98, 112, 126, 140, 154
21	Benzenepropanoic acid, α-hydroxy-, methyl ester	12,953–13,021	C_10_H_12_O_3_	43, 64, 76, 90, 102, 120, 161
22	1,3-Phenylene, bis(3-phenylpropenoate)	13,177–13,231	C_24_H_18_O_4_	43, 56, 76, 90, 102, 130, 161
23	1,2:4,5:9,10-Triepoxydecane	13,705–13,754	C_10_H_16_O_3_	43, 56, 68, 80, 92, 118, 206
24	1,3-Cyclohexadiene, 5-(1,5-dimethyl-4-hexenyl)-2-methyl-, [S-(R*,S*)]-	14,150–14,20	C_15_H_24_	40, 56, 70, 84, 92, 104, 118, 131, 203
25	1-Octadecanesulphonyl chloride	14,341–14,379	C_18_H_37_ClO_2_S	42, 70, 76, 92, 118, 140, 154, 201
26	Oxalic acid, allyl pentadecyl ester	15,079–15,125	C_20_H_36_O_4_	42, 56, 70, 84, 110
27	2-Piperidinone, N-[4-bromo-n-butyl]-	15,472–15,510	C_9_H_16_BrNO	43, 56, 70, 90, 104, 122, 206
28	2-Bromotetradecane	16,892–16,959	C_14_H_29_Br	42, 71, 98, 126, 154, 206
29	Oxalic acid, 6-ethyloct-3-yl heptyl ester	17,021–17,059	C_19_H_36_O_4_	42, 56, 84, 98, 110, 126, 206
30	2-Bromo dodecane	17,409–17,504	C_12_H_25_Br	42, 56, 70, 76, 84, 98, 112, 141, 168, 206
31	Ethylamine, 2-(adamantan-1-yl)-1-methyl-	17,991–18,037	C_13_H_23_N	43, 54, 68, 82, 106, 134, 206
32	Pentadecanal-	18,452–18,511	C_15_H_30_O	43, 56, 68, 81, 108, 206
33	Di-n-decylsulfone	18,662–18,735	C_20_H_42_O_2_S	43, 56, 70, 84, 98, 206
34	Dodecane, 1-iodo-	19,675–19,756	C_12_H_25_I	42, 70, 98, 112, 126, 154, 183, 206
35	Hexadecanoic acid, methyl ester	20,033–20,093	C_17_H_34_O_2_	42, 54, 73, 86, 96, 110,142, 170, 206, 227, 270
36	Benzenepropanoic acid, 3,5-bis(1,1-dimethylethyl)-4-hydroxy-,	20,136–20,200	C_18_H_28_O_3_	43, 56, 68, 90, 104, 116, 134, 146, 206
37	Methyl 3-bromo-1-adamantaneacetate	21,087–21,184	C_13_H_19_BrO_2_	43, 54, 66, 78, 90, 104, 116, 132, 156, 190, 206, 256, 280
38	1-Cyclohexylnonene	21,375–21,456	C_15_H_28_	43, 56, 68, 81, 95, 109, 132, 190, 206, 280
39	4-Methyl-2,4-bis(p-hydroxyphenyl)pent-1-ene, 2 TMS derivative	21,765–21,811	C_24_H_36_O_2_Si_2_	43, 54, 81, 104, 132, 158, 206, 280
40	Cyclohexene, 1-nonyl-	21,930–21,978	C_15_H_28_	40, 54, 66, 81, 95, 109, 123, 146, 190, 206, 280
41	Cyclotrisiloxane, hexamethyl-	22,121–22,194	C_6_H_18_O_3_Si_3_	43, 70, 84, 95, 114, 132, 190, 280
42	1,4-Bis(trimethylsilyl)benzene	22,245–22,309	C_12_H_22_Si_2_	43, 68, 84, 95, 118, 132, 146, 190, 206, 280
43	9,12,15-Octadecatrienoic acid, methyl ester, (Z,Z,Z)-	22,323–22,366	C_19_H_32_O_2_	43, 54, 66, 78, 94, 107, 120, 134, 148, 190, 280
44	Eicosane, 1-iodo-	22,501–22,608	C_20_H_41_I	42, 70, 85, 112, 126, 155, 190, 206, 238, 280
45	Arsenous acid, tris(trimethylsilyl) ester	24,536–24,585	C_9_H_27_AsO_3_Si_3_	43, 54, 74, 104, 132, 162, 190, 206, 236, 280, 314

* Note: TMS = trimethylsilyl.

**Table 2 metabolites-13-00281-t002:** Binding affinity of identified compounds to selected target proteins. Nd: not detected.

No	Name of Compounds	Binding Affinity between Ligands and Target Proteins (kcal/mol)			
		Antibacterial (3VSL)	Antioxidant (3EUB)	Antiplasmodial (3QS1)	Anticancer (3GD4)
1	Methane, dichloronitro-	−3.9	−4.4	−3.7	−4.2
2	Ether, hexyl pentyl:	−4.2	−4.8	−4.5	−4.9
3	1,5-Heptadien-4-ol, 3,3,6-trimethyl-	−5.4	−5.3	−5.0	−5.0
4	Sulfurous acid, 2-ethylhexyl isohexyl ester	−4.3	−4.9	−4.7	−4.5
5	Sulfurous acid, hexyl nonyl ester	−4.6	−5.3	−4.4	−4.7
6	Decane, 2,4-dimethyl-	−4.5	−4.4	−4.6	−4.7
7	Octane, 3,4,5,6-tetramethyl-	−5.2	−5.3	−4.7	−4.7
8	Oxalic acid, cyclohexyl nonyl ester	−5.4	−5.1	−4.0	−52
9	Octane, 2,4,6-trimethyl-	−5.3	−5.1	−4.7	−5.3
10	N-[β-Hydroxy-β-[4-[1-adamantyl-6,8-dichloro]quinolyl]ethyl]piperidine	−9.2	−10.8	−9.0	−9.3
11	1-Chloroundecane	−4.9	−5.2	−4.7	−5.1
12	Dodecane, 1-chloro-	−3.8	−5.0	−4.7	−5.4
13	Decane, 1-chloro-	−4.3	−4.8	−3.1	−4.4
14	Propanoic acid, 2,2-dimethyl-, 2-(1,1-dimethylethyl)phenyl ester	−6.3	−6.6	−6.0	−7.9
15	Sulfurous acid, decyl hexyl ester	−4.9	−4.7	−3.9	−4.6
16	Tetradecane, 1-chloro-	−4.2	−4.8	−3.7	−4.1
17	Dodecane, 4,6-dimethyl-	−4.3	−5.7	−4.6	−5.6
18	Cyclohexene, 2-ethenyl-1,3,3-trimethyl-	−5.3	−5.7	−5.2	−5.5
19	Pentane, 2,3,3-trimethyl-	−4.9	−4.7	−4.3	−4.8
20	Hexadecane	−4.0	−5.3	−3.8	−4.5
21	Benzenepropanoic acid,α-hydroxy-, methyl ester	−6.2	−6.6	−5.6	−5.9
22	1,3-Phenylene, bis(3-phenylpropenoate)	−9.4	−8.4	−8.7	−8.3
23	1,2:4,5:9,10-Triepoxydecane	−4.5	−5.7	−4.3	−5.5
24	1,3-Cyclohexadiene, 5-(1,5-dimethyl-4-hexenyl)-2-methyl-, [S-(R*,S*)]-	−5.7	−6.4	−7.4	−5.0
25	1-Octadecanesulphonyl chloride	−5.4	−4.5	−4.3	−6.6
26	Oxalic acid, allyl pentadecyl ester	−5.4	−4.8	−4.2	−5.2
27	2-Piperidinone, N-[4-bromo-n-butyl]-	−4.8	−4.9	−3.7	−5.7
28	2-Bromotetradecane	−4.0	−5.2	−4.2	−4.2
29	Oxalic acid, 6-ethyloct-3-yl heptyl ester	−5.3	−5.9	−5.4	−7.0
30	2-Bromo dodecane	−3.8	−5.3	−3.8	−4.4
31	Ethylamine, 2-(adamantan-1-yl)-1-methyl-	−6.0	−7.0	−6.6	−7.3
32	Pentadecanal-	−3.8	−5.3	−4.5	−5.0
33	Di-n-decylsulfone	−4.6	−4.6	−4.5	−5.0
34	Dodecane, 1-iodo-	−3.9	−4.8	−3.2	−5.0
35	Hexadecanoic acid, methyl ester	−5.0	−5.4	−3.7	−6.2
36	Benzenepropanoic acid, 3,5-bis(1,1-dimethylethyl)-4-hydroxy-, methyl ester	−6.6	−6.5	−7.0	−6.6
37	Methyl 3-bromo-1-adamantaneacetate	−6.3	−6.1	−6.8	−7.4
38	1-Cyclohexylnonene	−5.5	−6.2	−3.8	−6.6
39	4-Methyl-2,4-bis(p-hydroxyphenyl)pent-1-ene, 2TMS derivative	−6.9	−6.5	−7.1	−6.7
40	Cyclohexene, 1-nonyl-	−5.0	−5.1	−4.7	−5.4
41	Cyclotrisiloxane, hexamethyl-	−4.8	−4.9	−4.4	−5.6
42	1,4-Bis(trimethylsilyl)benzene	−5.1	−5.7	−4.8	−4.9
43	9,12,15-Octadecatrienoic acid, methyl ester, (Z,Z,Z)-	−5.3	−6.7	−4.7	−7.6
44	Eicosane, 1-iodo-	−4.8	−5.0	−2.9	−4.4
45	Arsenous acid, tris(trimethylsilyl) ester	ND	ND	ND	ND

**Table 3 metabolites-13-00281-t003:** Binding affinity of identified compounds against selected target proteins.

No		Native Ligand	Binding Energy with Native Ligand (kcal/mol)	Binding Energy with N-[β-Hydroxy-β-[4-[1-adamantyl-6,8-dichloro]quinolyl]ethyl]piperidine (kcal/mol)	Binding Energy with 1,3-Phenylene, bis(3-phenylpropenoate) (kcal/mol)
1	Antibacterial protein target: 3VSL	Cefotaxime	−7.6	−7.8	−8.3
2	Antioxidant protein target: 3EUB	Flavin adenine dinucleotide (FAD)	−13.0	−10.1	−10.6
		Phosphonic acidmono-(2-amino-5,6-dimercapto-4-oxo-3,7,8A,9,10,10A-hexahydro-4H-8-oxa-1,3,9,10-tetraaza-anthracen-7-ylmethyl)ester (MTE)	−9.9	−3.6	−7.8
3	Antiplasmodial protein target: 3QS1	(4R)-3-[(2S,3S)-3-{[(2,6-Dimethylphenoxy)acetyl]amino}-2-hydroxy-4-phenylbutanoyl]-N-[(1S,2R)-2-hydroxy-2,3-dihydro-1H-inden-1-yl]-5,5-dimethyl-1,3-thiazolidine-4-carboxamide (KNI-10006)	−10.5	−10.6	−9.2
4	Anticancer protein target: 3GD4	Flavin adenine dinucleotide (FAD)	−14.6	−9.1	−10.1
		Nicotinamide adenine dinucleotide (NAD)	−10.5	−11.1	−10.5

**Table 4 metabolites-13-00281-t004:** Drug likeness results of the identified compounds from *B. cernua.* Nd: not detected.

Molecule	MW	H-Bond		Fraction Csp3	Rotatable Bonds	MR	TPSA	Consensus Log P	ESOL Log S	ESOL Class	Violation				
		Acceptors	Donors								Lipinski	Ghose	Veber	Egan	Muegge
1	129.93	2	0	1.00	1	24.62	45.82	0.84	−1.49	Very soluble	0	3	0	0	2
2	172.31	1	0	1.00	9	56.08	9.23	3.66	−2.97	Soluble	0	0	0	0	2
3	154.25	1	1	0.60	3	50.14	20.23	2.59	−2.53	Soluble	0	1	0	0	2
4	294.49	3	0	1.00	12	86.78	54.74	3.66	−4.67	Moderately soluble	0	1	1	0	1
5	292.48	3	0	1.00	15	84.67	54.74	4.71	−4.75	Moderately soluble	0	1	1	0	1
6	170.33	0	0	1.00	7	59.80	0	4.78	−4.30	Moderately soluble	1	0	0	0	3
7	170.33	0	0	1.00	5	59.80	0	4.48	−4.07	Moderately soluble	1	0	0	0	3
8	298.42	4	0	0.88	12	84.29	52.60	4.31	−4.79	Moderately soluble	0	0	1	0	1
9	156.31	0	0	1.00	5	54.99	0	4.26	−3.82	Soluble	1	1	0	0	3
10	457.46	2	1	0.67	4	130.69	33.12	5.98	−8.06	Poorly soluble	1	2	0	1	1
11	190.75	0	0	1.00	9	59.79	0	4.75	−4.41	Moderately soluble	1	0	0	0	3
12	204.78	0	0	1.00	10	64.59	0	5.12	−4.77	Moderately soluble	1	0	0	0	2
13	176.73	0	0	1.00	8	54.98	0	4.37	−4.04	Moderately soluble	1	0	0	0	3
14	232.36	1	0	0.56	4	74.65	17.07	4.07	−4.19	Moderately soluble	0	0	0	0	1
15	306.50	3	0	1.00	16	89.47	54.74	5.18	−5.11	Moderately soluble	0	1	1	1	2
16	232.83	0	0	1.00	12	74.21	0	5.86	−5.49	Moderately soluble	1	1	1	1	2
17	198.39	0	0	1.00	9	69.41	0	5.50	−5.02	Moderately soluble	1	0	0	0	3
18	150.26	0	0	0.64	1	51.61	0	3.44	−2.9	Soluble	0	1	0	0	2
19	114.23	0	0	1.00	2	40.31	0	3.17	−2.82	Soluble	1	1	0	0	2
20	226.44	0	0	1.00	13	79.03	0	6.42	−5.60	Moderately soluble	1	1	1	1	2
21	180.2	3	1	0.30	4	48.28	46.53	1.43	−1.95	Very soluble	0	0	0	0	1
22	374.43	4	0	0.17	8	109.67	52.6	4.68	−5.06	Moderately soluble	1	0	0	0	1
23	184.23	3	0	1.00	6	47.10	37.59	1.68	−1.09	Very soluble	0	0	0	0	1
24	204.35	0	0	0.60	4	70.68	0	4.47	−4.10	Moderately soluble	1	0	0	0	2
25	353.00	2	0	1.00	17	102.40	42.52	6.68	−6.72	Poorly soluble	1	1	1	1	2
26	340.50	4	0	0.80	19	100.35	52.60	5.78	−5.89	Moderately soluble	0	0	1	0	2
27	234.13	1	0	0.89	4	58.14	20.31	2.17	−2.35	Soluble	0	0	0	0	0
28	277.28	0	0	1.00	11	77.28	0	5.83	−5.61	Moderately soluble	1	1	1	1	2
29	328.49	4	0	0.89	16	96.02	52.60	5.33	−5.46	Moderately soluble	0	0	1	0	2
30	249.23	0	0	1.00	9	67.67	0	5.10	−4.89	Moderately soluble	1	0	0	0	2
31	193.33	1	1	1.00	1	60.71	26.02	3.05	−3.13	Soluble	0	0	0	0	2
32	226.40	1	0	0.93	13	74.42	17.07	5.04	−4.49	Moderately soluble	0	0	1	0	2
33	346.61	2	0	1.00	18	107.22	42.52	6.68	−6.22	Poorly soluble	1	1	1	1	2
34	296.23	0	0	1.00	10	72.76	0	5.41	−5.68	Moderately soluble	1	0	0	0	2
35	270.45	2	0	0.94	15	85.12	26.30	5.54	−5.18	Moderately soluble	1	1	1	0	1
36	292.41	3	1	0.61	6	87.68	46.53	4.22	−4.5	Moderately soluble	0	0	0	0	0
37	287.19	2	0	0.92	3	67.20	26.30	3.28	−3.53	Soluble	0	0	0	0	0
38	208.38	0	0	0.87	7	71.63	0	5.43	−5.07	Moderately soluble	1	0	0	0	2
39	412.71	2	0	0.42	8	128.08	18.46	6.22	−7.76	Poorly soluble	1	1	0	1	1
40	208.38	0	0	0.87	8	71.63	0	5.43	−4.76	Moderately soluble	1	1	0	0	2
41	222.46	3	0	1.00	0	55.70	27.69	1.16	−3.12	Soluble	0	0	0	0	0
42	222.47	0	0	0.50	2	72.40	0	3.44	−4.89	Moderately soluble	0	0	0	0	2
43	292.42	2	0	0.63	14	93.31	26.30	nd	nd	nd	nd	nd	nd	nd	nd
44	408.44	0	0	1.00	18	111.22	0	8.36	−8.58	Poorly soluble	1	1	1	1	3
45	326.49	2	0	1.00	4	77.56	26.30	1.43	−4.64	Moderately soluble	0	0	0	0	0

**Table 5 metabolites-13-00281-t005:** Pharmacokinetic results of the identified compounds from *B. cernua*.

Molecule	Bioavailability Score	GI Absorption	BBB Permeant	P-gp Substrate	Cyp Inhibitor					Log Kp (cm/s)	Alerts		Likeness Violation	Synthetic Accessibility
					1A2	2C19	2C9	2D6	3A4		PAINS	BRENK		
1	0.55	High	Yes	No	No	No	No	No	No	−6.07	0	3	1	2.46
2	0.55	High	Yes	No	Yes	No	No	No	No	−4.36	0	0	3	2.22
3	0.55	High	Yes	No	No	No	No	No	No	−5.07	0	1	1	3.08
4	0.55	High	No	Yes	No	No	No	No	Yes	−3.82	0	0	2	4.31
5	0.55	High	No	No	No	No	Yes	No	Yes	−3.48	0	0	2	3.97
6	0.55	Low	No	No	No	No	Yes	No	No	−2.98	0	0	2	2.28
7	0.55	Low	No	Yes	No	No	No	No	No	−3.39	0	0	2	2.88
8	0.55	High	Yes	No	Yes	No	No	No	No	−3.74	0	2	2	3.00
9	0.55	Low	Yes	No	No	No	No	No	No	−3.48	0	0	2	2.44
10	0.55	Low	No	Yes	No	No	Yes	No	No	−3	0	0	2	5.58
11	0.55	Low	No	No	Yes	No	No	No	No	−2.98	0	1	3	2.58
12	0.55	Low	No	No	Yes	No	No	No	No	−2.68	0	1	3	2.69
13	0.55	Low	No	No	Yes	No	No	No	No	−3.28	0	1	3	2.48
14	0.55	High	Yes	No	No	No	No	Yes	No	−4.44	0	0	2	2.12
15	0.55	High	No	Yes	No	No	Yes	No	Yes	−3.18	0	0	2	4.08
16	0.55	Low	No	No	Yes	No	No	No	No	−2.08	0	1	3	2.90
17	0.55	Low	No	No	No	No	Yes	No	No	−2.38	0	0	3	2.76
18	0.55	Low	Yes	No	No	No	No	No	No	−4.72	0	0	2	3.35
19	0.55	Low	Yes	No	No	No	No	No	No	−4.28	0	0	2	1
20	0.55	Low	No	No	Yes	No	No	No	No	−1.80	0	0	3	2.26
21	0.55	High	Yes	No	No	No	No	No	No	−6.37	0	0	1	1.81
22	0.55	High	Yes	No	No	Yes	Yes	No	Yes	−4.9	0	3	3	3.77
23	0.55	High	Yes	No	No	No	No	No	No	−6.86	0	1	1	3.64
24	0.55	Low	No	No	No	Yes	Yes	No	No	−3.88	0	1	2	4.81
25	0.55	Low	No	No	Yes	No	Yes	No	No	−1.91	0	0	3	4.36
26	0.55	High	Yes	No	No	No	Yes	No	No	−2.52	0	3	2	3.43
27	0.55	High	Yes	No	No	No	No	No	No	−6.24	0	1	1	1.81
28	0.55	Low	No	No	No	No	Yes	No	No	−2.60	0	1	2	4.43
29	0.55	High	Yes	Yes	Yes	Yes	Yes	No	No	−3.07	0	2	2	3.73
30	0.55	Low	No	No	No	No	Yes	No	No	−3.20	0	1	3	4.22
31	0.55	High	Yes	No	No	No	No	No	No	−4.97	0	0	2	3.92
32	0.55	High	Yes	No	Yes	No	No	No	No	−3.06	0	1	3	2.15
33	0.55	Low	No	No	Yes	No	Yes	No	No	−2.31	0	0	2	4.16
34	0.55	Low	No	No	Yes	No	Yes	No	No	−2.85	0	2	2	3.39
35	0.55	High	Yes	No	Yes	No	No	No	No	−2.71	0	0	2	2.53
36	0.55	High	Yes	No	No	No	No	Yes	No	−4.66	0	0	1	2.13
37	0.55	High	Yes	No	No	No	Yes	No	No	−5.68	0	1	0	4.94
38	0.55	Low	No	No	Yes	No	No	No	No	−2.62	0	1	2	3.13
39	0.55	Low	No	Yes	No	No	No	Yes	No	−2.54	0	1	3	3.75
40	0.55	Low	No	No	Yes	No	No	No	No	−2.89	0	1	3	3.43
41	0.55	High	Yes	No	No	No	No	No	No	−5.52	0	1	1	4.34
42	0.55	Low	Yes	No	No	No	No	Yes	No	−3.73	0	1	2	3.66
43	nd	nd	nd	nd	nd	nd	nd	nd	nd	nd	nd	nd	nd	nd
44	0.55	Low	No	Yes	Yes	No	No	No	No	−0.46	0	2	3	4.28
45	0.55	High	Yes	Yes	No	No	No	No	No	−4.86	0	1	1	5.21

## Data Availability

The main article and Appendix A have already provided all the data.

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
