# Peer review of "Breynia cernua: Chemical Profiling of Volatile Compounds in the Stem Extract and Its Antioxidant, Antibacterial, Antiplasmodial and Anticancer Activity In Vitro and In Silico"

_metabolites, 2023, doi:10.3390/metabo13020281_

Round 1
Reviewer 1 Report (Previous Reviewer 1)
Manuscript is interesting and has scientific value and has been improved/completed. However, still some points should be explained before acceptance for publication.
1) I agree with Authors that GC-MS method after derivatization allows to detect also nonvolatile components; therefore, it is rather unexpected that no flavonoids, phenolic acids or other common secondary metabolites was found in methanol stem extract. According cited ref [8] different groups of secondary metabolites were found in Breynia genus. Maybe analytical procedure was not appropriate (lack of detail) or Authors just focused on volatile components (if yes, it should be clearly stated in title)
2) Lines 102-104: “Preliminary study …. " – lack of reference
3) 2.2. section – line 121: ref 21 is review article and did not provide the detailed methodology for TSM. Add more detail on derivatization.
4) “Established findings demonstrated about compounds identification of leaf extract of several species such as B. coronata, B. fruticosa, B. retusa, B. androgyna, B. glauca B. officinalis, B. rostrata, and B. vitis-idaea.” – unclear sentence
5) Table 1: m/z should be added. Why some of the compounds are underlined (add the explanation in footnote)?
6) What was final concentration of DMSO in analyzed samples? Pure DMSO? According numerous literature data DMSO is cytotoxic.
7) Figure 4. “Control: cancer cell without addition of B. cernua extract” – cells should be treated with solvent used for solubilization of extract
8) Figures 4-5: figure legend should be more detailed
Minor comments:
1) there are still some editorial errors, e.g. lack of space (line 113), lack of italic in some places: in vitro, in vivo, in silico.
2) „.beta.” „.alpha.”, in the name of compounds should be replaced by letter symbol (ß, α)
Author Response
Thank you very much for the reviewer's comment, correction and suggestion, we have provided point by point response to reviewer on the uploaded revised manuscript
Point by point response to Reviewer 1
|
Reviewer 1 |
|||
|
1 |
I agree with Authors that GC-MS method after derivatization allows to detect also nonvolatile components; therefore, it is rather unexpected that no flavonoids, phenolic acids or other common secondary metabolites was found in methanol stem extract. According cited ref [8] different groups of secondary metabolites were found in Breynia genus. Maybe analytical procedure was not appropriate (lack of detail) or Authors just focused on volatile components (if yes, it should be clearly stated in title) |
Thank you very much for the comment |
To make compounds in the extract more volatile and easier to be detected by using GCMS, trimethyl sylyl (TMS) derivatization method was employed. Unfortunately, based on GC-MS result this derivatization method failed to detect compound with many hydroxyl groups such as flavonoids and phenolic acids. Therefore, in the revised manuscript we are mention that our study was focus on volatile compound and it was stated in title. (please see line 2 and 265) |
|
2 |
Lines 102-104: “Preliminary study …. " – lack of reference |
Thank you very much for the correction |
Manuscript regarding with the GC-MS analysis and antioxidant and antibacterial activity of B. cernua leave was in preparation, we have mention that in manuscript (please see line 104) |
|
3 |
2.2. section – line 121: ref 21 is review article and did not provide the detailed methodology for TSM. Add more detail on derivatization. |
Thank you very much for the suggestion |
The detail methodology of TSM derivatization method has been added in the manuscript (please see line 120) |
|
4 |
“Established findings demonstrated about compounds identification of leaf extract of several species such as B. coronata, B. fruticosa, B. retusa, B. androgyna, B. glauca B. officinalis, B. rostrata, and B. vitis-idaea.” – unclear sentence |
Thank you very much for the comment |
The sentence has been corrected in the manuscript (please see line 243) |
|
5 |
Table 1: m/z should be added. Why some of the compounds are underlined (add the explanation in footnote)? |
Thank you very much for the suggestion and question |
m/z of each identified compound has been added and underline has been removed from Table 1 (please see line 361) |
|
6 |
What was final concentration of DMSO in analyzed samples? Pure DMSO? According numerous literature data DMSO is cytotoxic. |
Thank you very much for the question |
To prevent toxicity of DMSO, concentration of the solvent was used at 0.5% v/v. Detailed concentration of DMSO has been added in the manuscript (please see line 116, 138, 175, 193, 208, 420, 424) |
|
7 |
Figure 4. “Control: cancer cell without addition of B. cernua extract” – cells should be treated with solvent used for solubilization of extract |
Thank you very much for the correction |
Actually, the control is referring to cancer cell treated with 0.5% v/v DMSO. The note has been corrected in Figure 4 and 5 (please see line 419 and 424) |
|
8 |
Figures 4-5: figure legend should be more detailed |
Thank you very much for the correction |
Legend of figure 4 and 5 has been added with more detail explanation (please see line 417 and 423) |
|
9 |
there are still some editorial errors, e.g. lack of space (line 113), lack of italic in some places: in vitro, in vivo, in silico. |
Thank you very much for the correction |
The editorial errors have been corrected in the manuscript (please see line 4, 44, 45, 78, 88, 99, 302, 631) |
|
10 |
„.beta.” „.alpha.”, in the name of compounds should be replaced by letter symbol (ß, α) |
Thank you very much for the correction |
the name of compounds has been replaced by letter symbol (ß, α) (please see line 37, 250, 251, 299, 445, 456, 475, 484, 509, 523, 528 and Table 1-3) |

Reviewer 2 Report (Previous Reviewer 2)
I have no further comments.
Author Response
Thank you very much for the reviewer's comment, correction and suggestion, we have provided point by point response to reviewer on the uploaded revised manuscript
|
Reviewer 2 |
|||
|
1 |
English language and style are fine/minor spell check required |
Thank you very much for the comment |
Typos and spelling error has been corrected in the manuscript (please see line 33, 44, 78, 281, 288, 302, 335, 365, 378, 440, 455, 570, 571, 575, 576, 578, 582, 586, 589, 591, 596) |

Round 2
Reviewer 1 Report (Previous Reviewer 1)
Thanks for Authors for detailed response. Indeed, the manuscript has been corrected according to my suggestions.
I have only minor comments:
1) still some components in Table are underlined
2) N-[.β.-Hydroxy-.β.-[4-[1-adamantyl… - I think that dots are unnecessary in the name of compound (check all text)
Author Response
point-by-point response to the reviewer has been added in the attached cover letter

This manuscript is a resubmission of an earlier submission. The following is a list of the peer review reports and author responses from that submission.
Round 1
Reviewer 1 Report
Manuscript is interesting and has scientific value; however, some issue should be considered and explained before acceptance for publication.
1) Lack of detailed information on preparing the extract for biological tests. Line 100: „The solvent was removed by oven drying (70°C).” What solvent was used to solve the residues? Authors used term „methanol extract” in further description; however, methanol alone is cytotoxic.
2) Authors used GC-MS method but it allows only to detect only volatile components and plant extracts usually contain a lot of non-volatile secondary metabolites e.g. flavonoids, phenolic acids and others which are responsible for biological activity. Why was investigation focused on volatile metabolites? - – justification is needed in Introduction. Moreover, it should be clearly stated (line 91 and title) that only volatile compounds has been investigated
3) Why stem was used for investigation? – justification is needed in Introduction
4) 2.1. section: lack of drying condition (time, temperature), lack of weight the plant material taken for maceration
5) 2.3.2. section: more detail should be added
6) Table 1. % contribution of particular component in investigated samples should be added. Moreover, why some of the compounds are underlined?
7) Figure 1 is unnecessary. IC50 should be given in the text.
8) Table 2 is unnecessary. The description should be added to the text
9) Figure 5. What was used as control? Add information under the Figure.
Minor comment: there are a lot of editorial errors, e.g. line 29: “antioxidasnt”, line 323“startinf”, lack of italic in some places, unnecessary commas and dot (e.g. lines 76, 104) or bracket (e.g. line 109), unnecessary spaces and many, many others
Reviewer 2 Report
The manuscript entitled " Breynia cernua: Chemical Profiling of the Stem Extract and Its Antioxidant, Antibacterial, Antiplasmodial and Anticancer Activity In Vitro and In Silico " has been reviewed. The manuscript include questionable information regarding scientific methodology and identified metabolite using GC-MS and it is unsuitable for publication in the Journal Metabolites owing to the followings:
1. I think some of the identified compounds in Table 1 is doubtful, FOR EXAMPLE, N-[.beta.-Hydroxy-.beta.-[4-[1-adamantyl-6,8-dichloro]quinolyl]ethyl]piperidine, and the compounds 11-13, 25, 27, 30 and 44 it seems these compounds are synthetic and were not reported as natural metabolites from botanical source. Moreover, compounds 39, 42, 45 are produced from derivatization process although the authors did not mention any additional step before GC-MS analysis.
2. In addition, the experimental design is questionable. The authors should use LC-MS to investigate the secondary metabolites of Breynia cernua as it was not documented that this plant has volatile oil so GC-MS would be invalid for metabolite identification and rationalizing the claimed biological effects of the plant.
3. How the authors could identify the metabolites without comparing observed Kovats indices?
4. This manuscript lacks novelty. Unfortunately, there are many other studies that have looked at Breynia cernua antioxidant, antibacterial, antiplasmodial and anticancer effects. While this study tries to link biological activities to molecular docking, but the evidence for this finding is limited in context of GC-MS questionable identification.
Minor comments:
1. Line 30, 73-81: some Latin names like P. falciparum, M. roseus, B. cereus, B. subtilis, … need to add the abbreviation meaning to make clear by authors.
2. Abstract needs to include the numerical result and at least one line of conclusion.
3. Introduction references are old. Authors need to add several references at least from the last 3 years.
4. Line 42: accesability needs to be corrected.
5. Line 63-69: How the description regarding vascular bundle, stomata and epidermis is considered as morphological characteristics?
6. Line 75: Agrobacterium tumefaciens Streptococcus faecalis add comma between names.
7. Line 104: Mention the manufacturing country of GC-MS devise.
8. Line 117 (Then made a sample solution of 1000 ppm)…. Need to rewrite and author should make it clear in terms of sample concentration and standard used?
9. Line 175: RCSB PDB: [15] 3VSL [17], 3EUB [16], reference number 17 was mentioned in the text before number 16.
10. The reference list: has some titles in full UPPERCASE letters like POTENTIAL OF CLAMATION AREA OF RECOAL MINING SITES IN MEDICAL FIELD, while others are written with all capitalized words like Effects of Extracts of Selected Medicinal Plants upon Hepatic Oxidative Stress. Furthermore, some journal names are written in full without abbreviations like Journal of Pharmaceutical Analysis that indicate deviation from journal reference style.